# Efficient and Informative Laboratory Testing for Rapid Confirmation of H5N1 (Clade 2.3.4.4) High-Pathogenicity Avian Influenza Outbreaks in the United Kingdom

**DOI:** 10.3390/v15061344

**Published:** 2023-06-09

**Authors:** Marek J. Slomka, Scott M. Reid, Alexander M. P. Byrne, Vivien J. Coward, James Seekings, Jayne L. Cooper, Jacob Peers-Dent, Eric Agyeman-Dua, Dilhani de Silva, Rowena D. E. Hansen, Ashley C. Banyard, Ian H. Brown

**Affiliations:** Virology Department, Animal and Plant Health Agency (APHA-Weybridge), Woodham Lane, Addlestone KT15 3NB, UK

**Keywords:** clade 2.3.4.4, H5N1, high-pathogenicity avian influenza virus (HPAIV), real-time reverse-transcription (RRT)-PCR, infected premises (IP), swabs

## Abstract

During the early stages of the UK 2021-2022 H5N1 high-pathogenicity avian influenza virus (HPAIV) epizootic in commercial poultry, 12 infected premises (IPs) were confirmed by four real-time reverse-transcription–polymerase chain reaction (RRT)-PCRs, which identified the viral subtype and pathotype. An assessment was undertaken to evaluate whether a large sample throughput would challenge laboratory capacity during an exceptionally large epizootic; hence, assay performance across our test portfolio was investigated. Statistical analysis of RRT-PCR swab testing supported it to be focused on a three-test approach, featuring the matrix (M)-gene, H5 HPAIV-specific (H5-HP) and N1 RRT-PCRs, which was successfully assessed at 29 subsequent commercial IPs. The absence of nucleotide mismatches in the primer/probe binding regions for the M-gene and limited mismatches for the H5-HP RRT-PCR underlined their high sensitivity. Although less sensitive, the N1 RRT-PCR remained effective at flock level. The analyses also guided successful surveillance testing of apparently healthy commercial ducks from at-risk premises, with pools of five oropharyngeal swabs tested by the H5-HP RRT-PCR to exclude evidence of infection. Serological testing at anseriform H5N1 HPAIV outbreaks, together with quantitative comparisons of oropharyngeal and cloacal shedding, provided epidemiological information concerning the chronology of initial H5N1 HPAIV incursion and onward spread within an IP.

## 1. Introduction

Avian influenza viruses (AIVs) are a diverse group of economically important pathogens, classified according to their subtype, determined by their haemagglutinin (HA, H1–H16) and neuraminidase (NA, N1–N9) genes [1]. High-pathogenicity (HP)AIV outbreaks in galliform poultry caused by H5 and H7 subtypes are typified by a rapid onset of high morbidity and mortality [2]. Among HPAIVs, H5Nx viruses of the “goose/Guangdong” (GsGd) lineage [3] remain a major global concern and continue to evolve into new clades, among which clade 2.3.4.4 is currently significant [4]. Europe has experienced six clade 2.3.4.4 epizootics since 2014, with the five European incursions since autumn 2016 having been caused specifically by subclade 2.3.4.4b [5,6,7,8,9,10,11,12,13,14]. The clade 2.3.4.4 viruses are also referred to as H5Nx HPAIVs because of their propensity for NA segment reassortment [15].

The winter 2020–2021 subclade 2.3.4.4b epizootic in the UK was dominated by H5N8 cases in wild birds and poultry, but featured a minority of NA reassortments, including the H5N1 subtype [5]. Once the epizootic declined, there were several reports of H5N1 in wild birds during summer 2021 in Europe [16]. The earlier minority H5N1 subtype then emerged as the dominant European strain by autumn 2021 [5], with the UK experiencing its largest ever HPAIV incursion, with 985 wild bird cases plus 110 outbreaks in poultry and captive bird populations by April 2022 [14].

Since the turn of the century, real-time reverse-transcription (RRT)-PCR technology has gained primacy for the detection of infectious animal diseases caused by RNA viruses, encompassing notifiable avian diseases (NADs), including H5- and H7-subtype AIVs [17]. For statutory NAD investigations (known as “Report Cases” in the UK), swabs are collected from both the oropharyngeal (OP) and cloacal (C) cavities of birds at poultry premises under disease suspicion, as per internationally agreed guidelines [18]. For poultry, AIV RRT-PCR results are interpreted at the flock level, whereby generic AIV testing by RRT-PCRs, typically targeting the conserved matrix (M) gene, enables universal and sensitive detection of all AIV subtypes [19,20]. Such generic AIV RRT-PCRs are accompanied by parallel subtype-specific H5 and H7 AIV RRT-PCRs to detect these notifiable agents in both their low-pathogenicity (LP)AIV and HPAIV variations [21,22]. These tests have featured during previous European incursions of clade 2.3.4.4 H5Nx [23,24,25,26,27] and earlier GsGd lineage H5N1 HPAIVs [28,29,30]. NA-specific RRT-PCRs also enable rapid subtyping of clade 2.3.4.4 viruses [31,32].

In view of the rising number of H5N1 cases and their potential effects on both field and laboratory resources, the AIV RRT-PCR results were analysed during autumn 2021 to see whether a more refined testing approach could be confidently applied across all poultry production types. For the first time, the extensive 2021–2023 epizootic utilised the more recently validated AIV RRT-PCRs for generic M-gene detection [33] and confirmation of the H5 HPAIV pathotype [34]. The selected testing approach was further monitored in UK poultry through to March 2022 to ensure its efficacy for outbreak diagnosis, and has continued to be applied during the ongoing H5N1 subclade 2.3.4.4b epizootic in 2022–2023.

## 2. Materials and Methods

### 2.1. Disease Suspicion (Report Case) Testing during Late 2021 

Twelve H5N1 (subclade 2.3.4.4b) HPAIV poultry outbreaks, referred to as infected premises (IPs), were identified in the UK between 5 November and 5 December 2021, affecting chickens (seven layer and one broiler IP), turkeys (three IPs) and ducks (one IP) (IPs 1–12, Table 1). The 11 galliform IPs were commercial poultry holdings, with the domestic duck IP (IP5) included to provide the first anseriform IP of the autumn/winter 2021–2022 epizootic. These IPs were diagnosed promptly following initial disease suspicion based upon suspect clinical signs of NAD [35]. Sampling included swabbing (OP and C) of typically 20 birds per epidemiological unit (unless indicated otherwise, Table 1) at the suspect premises, providing 514 swabs across the 12 IPs, submitted for laboratory testing at APHA Weybridge. Swabs were individually placed and expressed in 1 mL brain heart infusion broth containing antibiotics (BHIB; [36]), and RNA was extracted robotically using a Universal Biorobot (Qiagen) [22] for testing by four AIV RRT-PCRs; namely the M-gene [33], the subtype-specific H5 and N1 RRT-PCRs [21,31], and the H5 HPAIV-specific RRT-PCR, herewith referred to as the H5-HP RRT-PCR [34]. In order to exclude other NAD agents, extracted swab RNA was also tested by H7 subtype-specific and avian paramyxovirus type-1 (APMV-1, also a surrogate for Newcastle disease virus [NDV]) RRT-PCRs [22,37]. Analyses of swabs from these 12 IPs guided a decision to optimise AIV RRT-PCR testing for subsequent Report Cases from early December 2021 onwards.

### 2.2. Subsequent Report Case Testing (5 December 2021–March 2022)

By early December 2021, by including other premises (captive birds and backyard poultry, not included in Table 1), the total of confirmed H5N1 HPAIV IPs had reached 33 in the UK [14]. The collection of OP and C swabs in Report Cases remained unchanged, but the number of AIV RRT-PCRs was reduced to three, namely the M-gene, H5-HP and N1 RRT-PCRs. Examples of the altered testing approach are presented for the diagnosis of a further 29 IPs (5 December 2021 and 24 March 2022), which included chickens (eight layer and three broiler IPs), turkeys (nine IPs), ducks (seven IPs) and pheasants (two IPs) (IPs 13–41, Table 1), providing a further 1476 OP and C swabs.

### 2.3. AIV RRT-PCR Interpretation and Statistical Analyses

Previous AIV RRT-PCR validation studies of these tests and earlier outbreak experiences had established Ct 36 as the positive threshold cut-off [38]. Therefore, any bird which registered a Ct less than or equal to 36 for either the OP, C or both swabs, by any of the AIV RRT-PCRs, was classified as infected. Negative swab samples were any samples registering within the range Ct 36.01–39.99, together with those which registered “No Ct”, shown in the relevant figures as Ct 40. Uninfected birds were those which registered negative results for both swabs by all AIV RRT-PCRs. The results were interpreted at the flock level, whereby detection of at least one infected bird resulted in the whole premises being considered as infected. Analyses of the AIV RRT-PCR results, which confirmed the 12 IPs (November to early December 2021) and subsequent 29 IPs (December 2021–March 2022), included mean Ct determination for each AIV RRT-PCR at each IP, with associated standard deviation and standard error of the mean. Regression analyses of the tests were performed at the first 12 IPs. To assess whether any significant differences between OP and C shedding occurred at an IP, the Wilcoxon matched-pairs signed-rank test was used to compare the Ct values obtained from the OP and C tracts (grouped for individual birds) for each AIV RRT-PCR. Statistical analyses were carried out with GraphPad (PRISM v7.03) software, with *p* < 0.0001 noted as being of high significance.

### 2.4. Surveillance Testing of Commercial Duck Premises Considered to Be at Risk of H5N1 HPAIV Infection: Swab Pooling and AIV RRT-PCR Testing

The UK H5N1 HPAIV 2021–2022 epizootic also required the testing of apparently healthy birds at predominantly commercial duck farms that were neighbouring or epidemiologically linked to the confirmed IPs, to provide assurance of freedom from infection prior to movement to slaughter. From December 2021 until June 2022, this surveillance testing affected 120 commercial duck premises. Surveillance was important to demonstrate freedom from infection, so OP swabs were typically collected from 60 birds at each epidemiological unit at these farms, providing a total of 24,720 swabs. The swabs were assembled as pools of five OP swabs in 1ml BHIB [39], thereby providing a total of 4944 pools. RNA was extracted robotically from each pool and tested by the H5-HP RRT-PCR.

### 2.5. Identification of Nucleotide Mismatches in the Primer/Probe Binding Regions of the Four AIV RRT-PCRs

For the 41 IPs (Table 1), a whole-genome sequence (WGS) was generated to define the genetic variation across the H5N1 genotypes that had occurred in the UK during the 2021–2022 epizootic [5,40]. All IPs in this study provided full H5N1 HPAIV sequences except IP19 (Appendix A). The M, H5 and N1 gene sequences from the 40 H5N1 HPAIVs were selected and aligned by the MEGAX software package (version 10.0.5; [41]), with the primer and probe binding sequences of the four AIV RRT-PCRs assessed for nucleotide mismatches. The four AIV RRT-PCRs were the M-gene, H5, H5-HP and N1 RRT-PCRs noted above, which featured in the testing of the first 12 IPs. The subsequent 29 IPs were tested by the M-gene, H5-HP and N1 RRT-PCRs. 

### 2.6. Serology Testing at Duck IPs

In addition to OP and C swabs, the 8 duck IPs (Table 1) provided 264 sera from live ducks. Duck sera were heat-inactivated at 56 °C for 30 min and then preabsorbed with chicken erythrocytes prior to testing for H5-specific antibodies by haemagglutination inhibition (HI) assays, which used three inactivated antigens (each at four haemagglutination units [18]), namely (i) an earlier but antigenically related clade 2.3.4.4 virus (A/duck/England/36254/2014 (H5N8) [36]), and two H5 antigens that served to detect H5-specific humoral responses to non-GsGd lineage H5 AIVs, namely (ii) A/teal/England/7394-2805/2006 (H5N3) and (iii) A/chicken/Scotland/1959 (H5N1) [42]. For APMV-1 antibody responses, the chicken/Ireland/Ulster/1967 isolate served as the inactivated HI antigen. The inactivated H7N7 antigen (A/turkey/England/647/1977; [42]) excluded evidence of past H7 AIV infection in the duck flocks.

## 3. Results

### 3.1. AIV RRT-PCR Testing of 12 IPs (Autumn 2021): Four-Test Strategy

The first 12 UK commercial IPs diagnosed during November and early December 2021 as being infected by H5N1 HPAIV involved testing by the M-gene, H5, H5-HP and N1 RRT-PCRs. The 514 swabs (OP and C) from 257 birds included 153 layers (seven IPs), 22 broilers (one IP), 62 turkeys (three IPs) and 20 ducks (one IP), with Ct value distributions shown for the four tests (Figure 1). Comparison of the mean Ct values for the assays showed that the H5-HP RRT-PCR was the most sensitive, followed by the H5, M-gene and N1 RRT-PCRs (Figure 1).

Regression analyses of paired tests showed that the M-gene, H5 and H5-HP RRT-PCRs were closely equivalent to each other, which was suggestive of linearity across their diagnostic range (i.e., ≤36.0), as revealed by a slope that approached unity, i.e., >0.917 (Appendix A). Less equivalence was discernible when the N1 RRT-PCR was compared to the other three tests where the slope ranged from 0.801–0.845 (Appendix A), with the regression lines displaying greater intercept values on the vertical axes, which corroborated the reduced sensitivity of the N1 RRT-PCR compared to the three other tests (Figure 1).

Between 96–100% of the sampled birds were infected at all 12 IPs (Figure 2 and Figure 3, Table 1). All four AIV RRT-PCRs were fit-for-purpose at the flock level in the three poultry species, as evidenced by successful and sensitive generic AIV detection together with correct subtyping (HA and NA) and pathotyping as H5N1 HPAIV. In addition, H7 and L-gene (APMV-1/NDV) RRT-PCR testing excluded the presence of other NAD pathogens. However, to streamline the diagnostic testing, in early December 2021, it was decided to cease routine use of the H5 RRT-PCR. Therefore, the more sensitive H5-HP RRT-PCR rendered the H5 RRT-PCR to be superfluous for outbreak confirmation due to this particular H5N1 subclade 2.3.4.4b strain. The H5-HP RRT-PCR only amplifies H5 HPAIVs; hence, rapid molecular confirmation of this pathotype [34] enabled immediate implementation of legislated HPAI control measures. The M-gene RRT-PCR was retained in the event of any AIV incursions caused by different subtypes, to reflect consistency with international standards. The absence of H7 AIV circulation across the UK and Europe meant that the H7 RRT-PCR was also removed from frontline testing. Despite some sensitivity concerns regarding the N1 RRT-PCR, it remained effective at the flock level in confirming the NA(N1) subtype, and was therefore retained for the three-test approach. The L-gene RRT-PCR was also retained for Report Case testing to detect any NAD incursions due to APMV-1, as a surrogate for NDV.

The need to test both OP and C swabs with their continuing impact on outbreak resources was also considered. Among the same 12 IPs, the distribution of AIV RRT-PCR Ct values was analysed separately for OP and C swabs, for each of the four assays (Figure 2 and Figure 3). Overall, for each AIV RRT-PCR, there was a trend for OP swabs to yield stronger Ct values than C swabs collected from the same bird. A highly significant difference (*p* < 0.0001) for OP swabs to be more sensitive for swabbing was apparent for all four tests at only two of these IPs, namely one turkey IP (Figure 3a) and one duck IP (Figure 3d). Six IPs included differences between OP and C shedding (albeit at weaker significance), while three chicken IPs (Figure 2b,g,h) and one turkey IP (Figure 3b) revealed no significant difference between OP and C shedding, as determined by all four tests. Therefore, it was considered prudent to continue collecting both OP and C swabs during Report Case investigations. 

### 3.2. AIV RRT-PCR Testing of 29 Subsequent IPs (December 2021–March 2022): Three-Test Strategy

To monitor how the three-test approach performed during the subsequent months of the 2021–2022 UK H5N1 epizootic, the AIV RRT-PCR results from a further 29 IPs were analysed (December 2021–March 2022; Figure 4). These example IPs represented different poultry sectors (IPs 13–41; Table 1) and provided a further 1476 OP and C swabs for analysis. The H5-HP RRT-PCR remained the most sensitive assay overall, with its mean Ct value slightly stronger than those of the M-gene RRT-PCR, followed by the N1 RRT-PCR (Figure 4a). The same sensitivity hierarchy was apparent for the layer, broiler and pheasant sectors (Figure 4b,c,f respectively), thus affirming the three-test approach as effective in identifying H5N1 HPAIV-affected IPs. 

All Report Cases described in this study originated from initial observations of clinical signs in poultry, with the exception of two commercial duck IPs, which will be described in more detail in Results Section 3.3. At 24 of the 29 additional IPs, high infection prevalence (>90%) among the sampled birds reflected rapid H5N1 spread within these farms. Among the remaining five IPs, lower infection prevalences were observed at layer IP19 (60%), broiler IP26 (23%) and three duck farms (IPs 39, 40 and 41, each at 73% prevalence; Table 1); hence, the three-test approach was also effective at lower prevalences. IPs 19, 39, 40 and 41 were sampled at two epidemiological units, where differing shedding prevalences suggested incursion into one unit ahead of the other. 

In comparing OP and C shedding at these 29 IPs (Appendix A), the trend was again for OP shedding to be stronger than C shedding at 28 IPs. Highly significant (*p* < 0.0001) shedding differences by all three AIV RRT-PCRs occurred at two of eight layer IPs (Appendix A), one of three broiler IPs (Appendix A), six of nine turkey IPs (Appendix A) and four of seven duck IPs (Appendix A). The absence of any significant difference between OP and C shedding by all three AIV RRT-PCRs occurred at seven of twenty-nine IPs, namely four of eight layer IPs (Appendix A), two of three broiler IPs (one of which was the only instance of mean C shedding being stronger than mean OP shedding; Appendix A) and one of two pheasant IPs (Appendix A).

Duck IP40 was sampled at two combined epidemiological units (Table 1 and Appendix A) with a considerable difference in individual shedding prevalence at 100% and 40% (Appendix A). Interestingly, IP40 represented the only duck premises in this study that demonstrated H5 seropositivity by HI, which included clade 2.3.4.4 H5N8 antigen testing (Appendix A). In addition, 7/24 (29%) sampled ducks in the first unit at IP40 were coinfected with APMV-1, as evidenced by simultaneous shedding (but no seroconversion), while no APMV-1 infection was detected in the second unit (Appendix A).

### 3.3. Surveillance by AIV RRT-PCR at Commercial Duck Premises That Represented Further H5N1 HPAIV Infection Risk: IPs 39 and 41

There was also a need to test apparently healthy commercial ducks at neighbouring (e.g., in disease control zones) or epidemiologically linked premises that were considered to be at risk for infections. A focused laboratory strategy featured sampling and testing of 60 ducks at each epidemiological unit, with only OP swabs collected as pools of five. This approach enabled efficient handling at reduced cost without impacting the diagnostic outcomes. RNA extracted from the OP swab pools was tested only by the H5-HP RRT-PCR by virtue of its high sensitivity. The decision to apply this surveillance approach was taken during December 2021, at which time the duck IPs had revealed a highly significant (*p* < 0.0001) difference between OP and C shedding at three outbreaks (IPs 5 (Figure 3d), 22 and 27 (Appendix A). This surveillance approach identified H5N1 infection at the pre-clinical stage at two at-risk duck premises (Appendix A). Subsequent surveillance at one duck premises revealed two houses to include 1/12 and 12/12 OP pools as positive by H5-HP RRT-PCR. At the second duck premises, one epidemiological unit was similarly assessed with 1/12 pools as positive by H5-HP RRT-PCR (Appendix A). These results immediately prompted Report Case sampling at both premises, leading to confirmation of infection, namely IPs 39 and 41 (Table 1, Appendix A).

### 3.4. Assessment of Nucleotide Mismatches at the Primer/Probe Binding Sequences Identified during the H5N1 HPAIV UK Epizootic

H5N1 WGS data were generated at 40 of the IPs investigated in this study. Nucleotide sequence alignments were carried out for the primer/probe binding regions that featured in infection confirmation at IPs 1–12, with similar alignments prepared for the three AIV RRT-PCRs that featured in infection confirmation at IPs 13–41, with the exception of IP 19, where WGS was unavailable (Figure 5). For the M-gene RRT-PCR, the primer/probe binding sequences remained unchanged and aligned perfectly at all nucleotide positions for all 40 H5N1 sequences obtained between November 2021 and March 2022; hence, they are excluded from Figure 5. For the 40 WGSs, the H5, H5-HP and N1R RT-PCRs all essentially displayed 2 conserved patterns of nucleotide changes at their primer/probe binding sequences (Figure 5) which was associated with the H5N1 genotype (Table 1). The nucleotide mismatches appear to have occurred at noncritical positions in the primer/probe sequences, as there was no evidence for any marked compromise in the sensitivity of the AIV RRT-PCRs in detecting H5N1 infection during this epizootic, at least at the flock level. Additional nucleotide changes were occasionally observed in a limited number of H5N1 isolates (Figure 5), but these also did not appear to compromise the sensitivity of the AIV RRT-PCRs. 

## 4. Discussion

By early December 2021, the growing number of H5N1 subclade 2.3.4.4b HPAIV wild bird cases (235 confirmed in the UK since late October 2021, plus the 33 initial UK IPs, including the first 12 largely commercial IPs listed in Table 1 [43]) raised questions regarding the most efficient laboratory testing strategy during an escalating epizootic. Therefore, laboratory testing approaches were reviewed, particularly as the H5-HP RRT-PCR [34] was being used prospectively for the first time during a UK AIV epizootic. Analysis of the Ct values registered by the four AIV RRT-PCRs demonstrated their fitness for purpose, but informed the decision to cease using the H5 RRT-PCR for Report Case testing during the ongoing H5N1 HPAIV outbreaks during 2021–2022. Replacement by a three-test strategy exploited the advantages of the sensitive H5-HP RRT-PCR, which confirmed the HPAI pathotype. This test negated the need for the previously established, but more time-consuming, molecular pathotyping approach of first generating H5-specific amplicons that span the HA cleavage-site region [44,45], followed by their purification and sequencing [46]. In addition to saving on laboratory resources, the importance of time saving during confirmatory H5N1 HPAIV diagnosis cannot be underestimated. Previous European H5Nx clade 2.3.4.4 incursions have included NA gene reassortments [5,11]; hence, the continued use of the N1 RRT-PCR affirmed continued circulation of the H5N1 subtype during the epizootic.

Prompt and robust outbreak confirmation informed the rapid imposition of legislative NAD control measures. Consequently, temporary control zone impositions were avoided, which otherwise would have led to a greater burden for the licensure of poultry products. The H5-HP RRT-PCR also enabled implementation of the 3Rs principles [47], with no need to confirm the pathotype by the statutory in vivo approach, namely the intravenous pathogenicity index test [18]. There was a trend for OP shedding to be stronger than C shedding for all the poultry species. This observation inferred a greater viral tropism for the respiratory compared to the enteric tract. The statistical analysis was assessed at a high stringency across 41 IPs, which represented all the affected UK poultry sectors. However, the highly significant difference (*p* < 0.0001) for stronger OP shedding by the three selected AIV RRT-PCRs was observed at 15/41 (37%) IPs (Table 1). The absence of significant differences between OP and C shedding, as assessed by the three selected tests, was most apparent for the 15 layer IPs (Figure 2 and Appendix A). Only two layer IPs revealed highly significant differences for stronger OP shedding, but seven of the layer IPs showed no significant differences. This sector was particularly affected by H5N1 HPAIV in the UK, with 22 commercial layer IPs identified between November 2021 and January 2022 [14].

Experimental (synchronous) H5N8 chicken and turkey infections showed that initial shedding may occur a day earlier and at a higher mean titre from the OP compared to the C tract [48,49]. Outbreak dynamics differ by not only involving an unknown entry point for infection with an undefined viral dose, but also feature nonsynchronous dissemination within and sometimes between epidemiological units on a scale that cannot be replicated experimentally. The random sampling of the 31 nonsynchronously H5N1-infected commercial chicken and turkey IPs included only one broiler IP where mean C shedding was slightly greater than OP shedding, although the difference was not significant (Appendix A). Effective AIV RRT-PCR approaches are crucial to detect early viral shedding at low infection prevalence, as observed in broilers at IP26 (Table 1 and Appendix A). A universal change to exclusively OP swabbing for all AIV incursions in chickens would potentially reduce test sensitivity, particularly as successful AIV RRT-PCR diagnosis of earlier UK H5 and H7 LPAIVs in broiler breeders included smaller proportions of chickens actively shedding from the C tract only, which appeared to reflect the timing of sampling [50].

HPAIV detection in the two most important UK commercial galliform species (chickens and turkeys) is dependent on passive surveillance through the initial reporting of suspect clinical signs [35,51]. Experimental studies have demonstrated that rapid mortality occurs following the synchronous infection of chickens with earlier clade 2.3.4.4 and subsequent subclade 2.3.4.4b H5Nx HPAIV isolates [48,52], typically by 2 days post-infection (dpi), with shedding detectable at 1 dpi. Contact transmission investigations of an earlier UK H5N8 HPAIV among turkeys showed the mean time to death (MDT) to occur between 4.4–5.5 days post-contact (dpc) [49]. While high H5N8 mortality for this species [36] again underlines the importance of passive surveillance in commercial galliforms, a longer MDT in turkeys compared to chickens was observed with an earlier North American clade 2.3.4.4 H5N2 HPAIV [53], therefore again emphasising the importance of rapid laboratory diagnosis of suspect galliform outbreaks, including sensitive diagnosis at the preclinical stage. H5N1 HPAIV (GsGd clade 2.2) incursion was detected at low prevalence at the preclinical stage during surveillance at a turkey premises that was an at-risk contact following an earlier UK H5N1 HPAIV outbreak in 2007 [38]. The collection of both OP and C swabs should continue at suspect premises in accord with established principles [18].

For the eight domestic duck IPs diagnosed between November 2021 and March 2022, four revealed greater OP shedding at high significance, as determined by the AIV RRT-PCRs (Figure 3d and Appendix A). The four remaining duck IPs demonstrated differences at lower significance between OP and C shedding. Experimental duck infections with European H5Nx HPAIVs again showed that initial OP shedding tended to be greater than C shedding [36,48,49,52,54]. These studies also affirmed variable mortality in ducks, which can potentially compromise passive surveillance in farmed anseriforms. The first UK clade 2.3.4.4 (H5N8) HPAIV incursion during November 2014 at a duck breeder farm was suspected from clinical presentation, where secondary microbial infection likely contributed to the apparent virulence [55]. Therefore, the importance of accurate laboratory diagnosis for commercial ducks cannot be understated; hence, the collection of both OP and C swabs should continue. The high proportion of anseriform farms affected by the subsequent 2016–2017 European H5N8 epizootic [6] underlined the importance of sensitive laboratory diagnosis in this poultry sector. A crucial underlying aim remains the prevention of H5Nx GsGd HPAIV endemicity in UK and European poultry, particularly among farmed anseriforms, where the establishment of an insidious infection reservoir remains a threat, as observed earlier in East Asia [56].

The value of including appropriate serological testing for possible clade 2.3.4.4 Report Cases at commercial anseriform premises has been noted previously [36,57,58]. The current study included one duck premises (IP40) with H5 seroconversion, which emphasised the importance of using an immunologically matched clade 2.3.4.4 antigen for optimal H5 HI testing (Appendix A). These ducks were sampled at a time post-incursion which enabled detection of clear H5 seroconversion. The relatively high seroconversion, together with the low prevalence of ongoing H5N1 shedding at low titres at the second unit, suggested that sampling had occurred at a comparatively late stage following H5N1 incursion (Appendix A), with infection beginning to be resolved in these ducks. Interestingly, IP40 also revealed cocirculation of H5N1 HPAIV and APMV-1 (avirulent, data not shown) within the first duck unit (Table 1 and Appendix A). Experimental coinfection of ducks with NDV and an earlier H5N1 GsGd HPAIV isolate resulted in mutual replication interference for both viruses [59], but both the avirulent APMV-1 and H5N1 HPAIV were detected successfully by the RRT-PCR testing strategy. 

Regarding the two commercial pheasant IPs, the mean OP shedding tended to be stronger than the C shedding by the AIV RRT-PCRs, but no significant difference was observed at IP37 (Table 1). Clinical signs similarly alerted previous European subclade 2.3.4.4b HPAIV incursions in both free and farmed pheasants [60,61,62], which has been confirmed experimentally [63]. Gamebirds attract interest as potential bridging hosts between wild birds and farmed poultry such as chickens [64], with affected farms also demanding prompt outbreak interventions.

The control and management of HPAIV poultry incursions are informed by an effective combination of surveillance allied to rapid and sensitive laboratory diagnosis, where validated AIV RRT-PCRs remain crucial for confirming infection [65]. However, the study also demonstrated the additional importance of assessing fitness for purpose whenever a novel AIV incursion occurs. This information enabled a confident decision to be made in early December 2021 to modify the testing strategy, thereby improving the efficiency of the laboratory testing process when dealing with a large epizootic. While observing the internationally mandated recommendations for notifiable AIV outbreak interventions, this study showed how reference laboratories can apply ongoing outbreak data to inform a successful and sensitive testing strategy for a given AIV strain within different poultry sectors, leading to effective disease control, interventions and management [66].

Examination of the AIV RRT-PCR primer/probe binding sequences from UK H5N1 isolates demonstrated that any nucleotide mismatches did not compromise assay robustness in terms of test sensitivity, as demonstrated at the flock level during the epizootic period between November 2021 and March 2022. Although the majority of UK commercial IPs experienced infection at high H5N1 prevalence, several individual premises and epidemiological units were successfully diagnosed at lower prevalence. This approach was extended to the successful surveillance of apparently healthy ducks, where the novel H5-HP RRT-PCR [34] and the earlier validated pooling strategy [39] successfully led to prompt outbreak diagnoses at IPs 39 and 41.

The H5N1 subclade 2.3.4.4b HPAIV had also spread to North America by late 2021 [67], with subsequent wild bird and poultry cases reported in 2022. At the time of writing, recurrent H5N1 HPAIV UK and European outbreaks had revived during autumn 2022 and continued during the ensuing winter [68], underlining that H5Nx subclade 2.3.4.4b HPAIVs represent a continuing global epizootic threat, along with its potential for zoonotic infections [69]. 

## Figures and Tables

**Figure 1 viruses-15-01344-f001:**
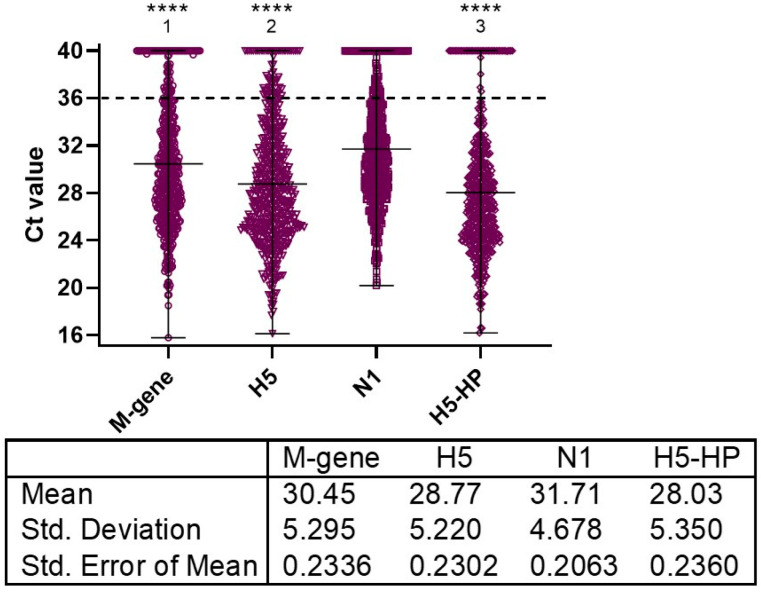
Distribution of Ct values obtained by the 4 AIV RRT-PCRs during diagnosis of the 12 poultry IPs during the early stage (November–early December 2021) of the 2021–2022 H5N1 HPAIV epizootic in the UK. The IPs included commercial layers (*n* = 7), commercial broiler breeders (*n* = 1), commercial turkeys (*n* = 3) and domestic ducks (*n* = 1), which provided 514 OP and C swabs from 257 birds (listed as IPs 1–12 in Table 1), with statistical descriptors tabulated for the datasets. The Ct values obtained from each RRT-PCR assay were compared using a Friedman test followed by Dunn’s multiple comparisons test. Statistically significant differences between the RRT-PCR Ct values are shown (****, *p* < 0.0001) for the following comparisons: ^1^ compared to N1 only; ^2^ compared to M-gene and N1; and ^3^ compared to H5, M-gene and N1.

**Figure 2 viruses-15-01344-f002:**
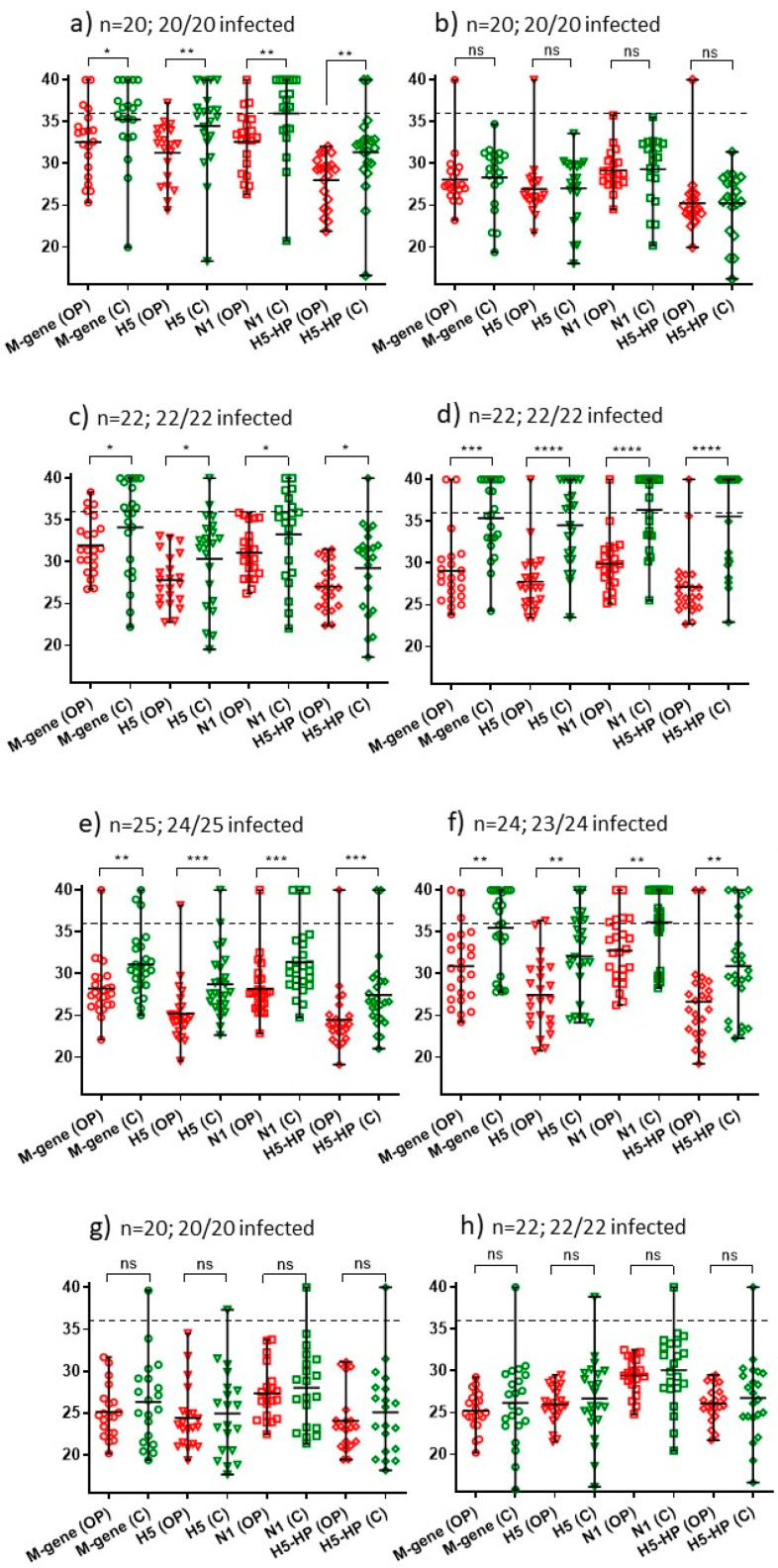
Distribution of Ct values as a measure OP and C shedding at eight commercial chicken IPs identified during November and early December 2021. Panels (**a**–**g**) show the Ct values for the layer IPs 3, 6, 7, 8, 10, 11 and 12, respectively; panel (**h**) shows the Ct values from IP9 (broiler breeders) (Table 1). Ct values were registered by four AIV RRT-PCRs, with the mean and range of Ct values indicated. Significant differences between the stronger OP compared to C shedding are indicated by **** (*p* < 0.0001, high significance), *** (*p* < 0.001), ** (*p* < 0.01), * (*p* < 0.05) and ns (not significant). The proportions of H5N1 HPAIV-infected birds among those which were swabbed are indicated for each IP (also summarised in Table 1).

**Figure 3 viruses-15-01344-f003:**
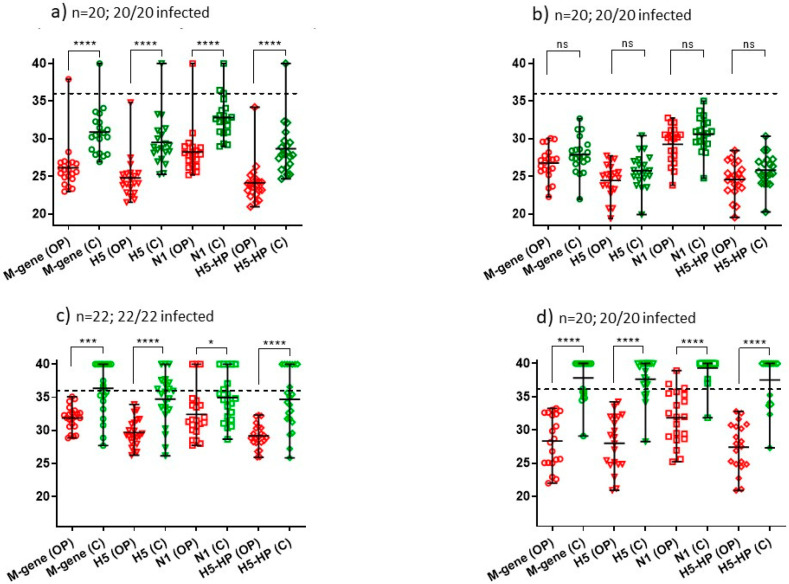
Distribution of Ct values as a measure of OP and C shedding at three commercial turkey IPs (IPs 1, 2 and 4 (Table 1) in panels (**a**–**c**) respectively) and domestic duck IP 5 (panel (**d**)) identified during November and early December 2021. Ct values were registered by four AIV RRT-PCRs. See Figure 2 legend for other details.

**Figure 4 viruses-15-01344-f004:**
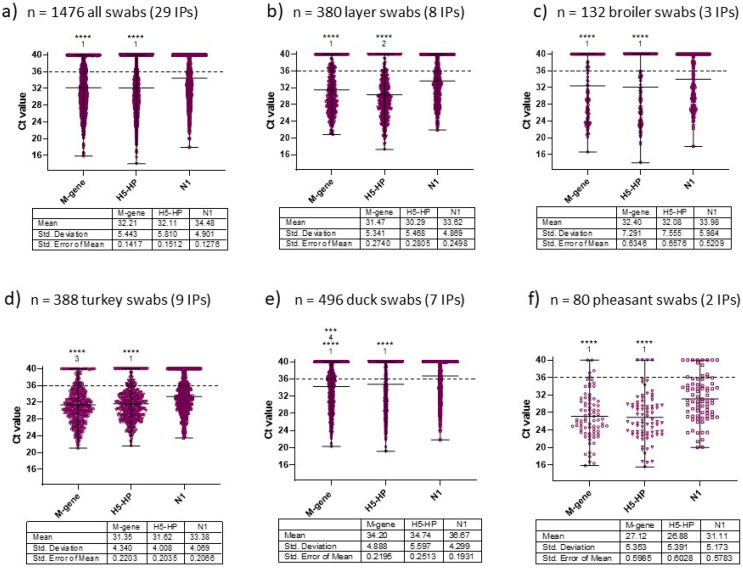
Distribution of Ct values obtained at an additional 29 poultry IPs by the 3 RRT–PCRs, confirmed between early December 2021 and April 2022. A total of 1476 OP and C swabs were collected from (**a**) all 738 birds, with division of the 29 IPs according to the poultry sectors, namely (**b**) layers (*n* = 8), (**c**) broilers (*n* = 3), (**d**) turkeys (*n* = 9), (**e**) ducks (*n* = 7) and (**f**) pheasants (*n* = 2). These birds are listed as IPs 13–41 (Table 1). Statistical descriptors for each dataset are tabulated for each panel. The Ct values obtained from each RRT-PCR assay were compared using a Friedman test followed by a Dunn’s multiple comparisons test. Statistically significant differences between the RRT-PCR Ct values are shown (***, *p* < 0.001; ****, *p* < 0.0001) for the following comparisons: ^1^ compared to N1 only; ^2^ compared to M-gene and N1; ^3^ compared to H5-HP and N1; ^4^ compared to H5–HP only.

**Figure 5 viruses-15-01344-f005:**
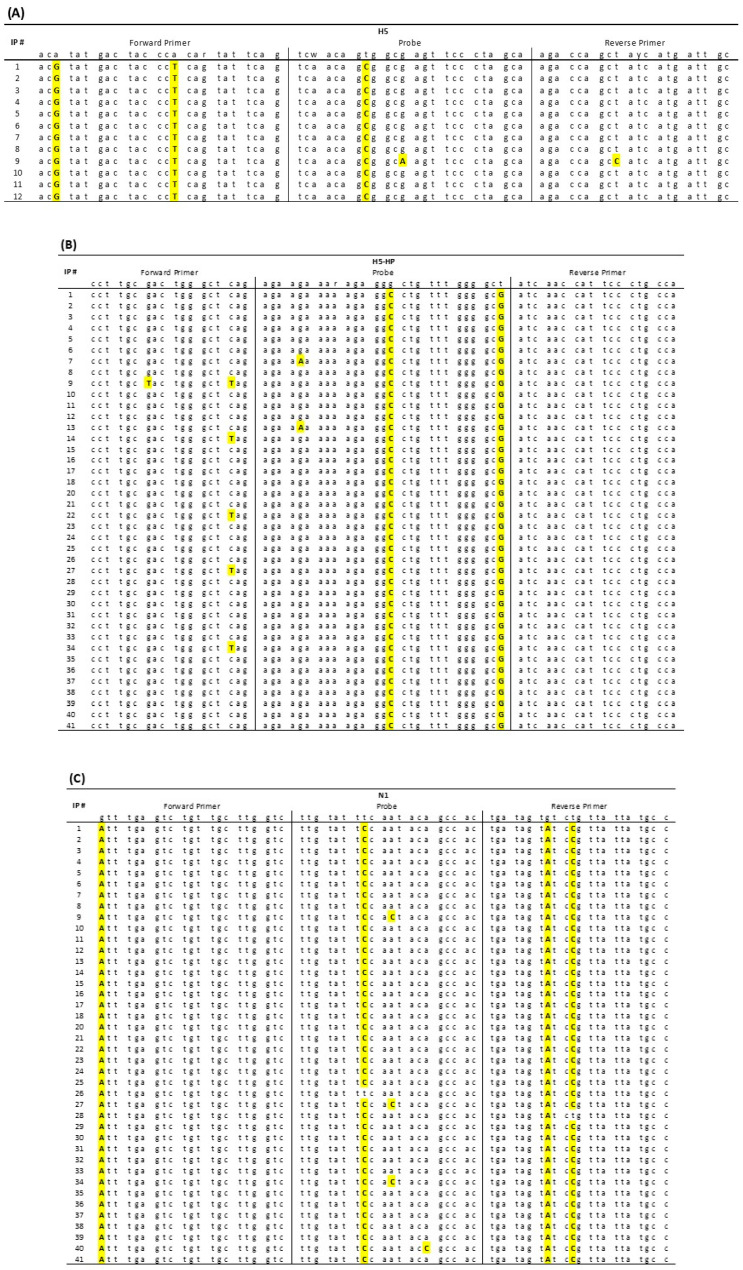
Nucleotide sequence alignments focusing on the primer and probe binding sequences of the (**A**) H5, (**B**) H5-HP and (**C**) N1 RRT-PCRs that were used to confirm H5N1 HPAIV incursion at the indicated IPs. WGS data were obtained from 40 of the 41 IPs (i.e., all except IP19) affected by H5N1 HPAIV outbreaks, with yellow highlights indicating nucleotide sequence mismatches. The header of each panel includes the corresponding primer and probe sequences for each test, written 5′ to 3′, left to right. No nucleotide sequence mismatches were observed for the M-gene RRT-PCR primers and probe at any of the 40 IPs.

**Table 1 viruses-15-01344-t001:** Forty-one poultry IPs affected by H5N1 HPAIV infection between November 2021 and March 2022 in the UK. The first 12 IPs (5 November 2021–5 December 2021) are listed chronologically, followed by a further 29 example IPs (December 2021–March 2022) diagnosed by the four- and three-test approaches, respectively. “Report Case date” indicates when the decision was made to submit clinical samples (including the swabs analysed in this study) to APHA Weybridge, based on suspect clinical presentation, except for IPs 39 and 41 (commercial ducks), where Report Cases were declared following premovement testing (Appendix A). AIV RRT-PCR testing was typically completed during the following day, and the H5N1 genotype was as defined by Pohlmann et al. (2022). Significant differences between the stronger OP compared to C shedding are indicated by **** (*p* < 0.0001, high significance), *** (*p* < 0.001), ** (*p* < 0.01), * (*p* < 0.05) and ns (not significant). Stronger mean C shedding compared to OP shedding was observed at only one farm, namely the broiler breeder IP 24, indicated by *italic text*.

Infected Premises (IP) ID in Chrono-Logical Order and Report Case Date	Poultry Sector: Numbers of Birds Kept across the Whole IP	Number of Epidemiological Units Sampled Per IP; Total Numbers of Swabbed (OP and C Swabs) Birds; Proportion of Infected Birds among Those Sampled (Percentage in Parentheses), with Proportions Per Unit in Square Parentheses, Where Applicable	Significant Difference between OP and C Shedding for the AIV RRT-PCRs among the Swabbed Birds (Figure Reference in Parentheses)	H5N1 Genotype
1 5/11/21	Small free-range turkey flock: 51	One; 20 pairs of swabs; 20/20 (100%)	M-gene: **** H5: **** N1: **** H5-HP: **** (Figure 3a)	B2
2 11/11/21	Commercial turkey rearers: 1400	One; 20 pairs of swabs; 20/20 (100%)	M-gene: ns H5: ns N1: ns H5-HP: ns (Figure 3b)	B2
3 11/11/21	Free-range layers: 120,000	One; 20 pairs of swabs; 20/20 (100%)	M-gene: * H5: ** N1: ** H5-HP: ** (Figure 2a)	B2
4 16/11/21	Commercial turkey rearers: 17,100	One; 22 pairs of swabs; 22/22 (100%)	M-gene: *** H5: **** N1: * H5-HP: **** (Figure 3c)	B2
5 17/11/21	Domestic ducks: 40	One; 20 pairs of swabs; 20/20 (100%)	M-gene: **** H5: **** N1: **** H5-HP: **** (Figure 3d)	B2
6 17/11/21	Commercial free-range layers: 12,000	One; 20 pairs of swabs; 20/20 (100%)	M-gene: ns H5: ns N1: ns H5-HP: ns (Figure 2b)	B2
7 29/11/21	Free-range layer chickens: 32,000	One; 22 pairs of swabs; 22/22 (100%)	M-gene: * H5: * N1: * H5-HP: * (Figure 2c)	B2
8 1/12/21	Commercial free-range layers: 16,000	One; 22 pairs of swabs; 22/22 (100%)	M-gene: *** H5: **** N1: **** H5-HP: **** (Figure 2d)	B2
9 1/12/21	Commercial broiler breeders: 26,000	One; 22 pairs of swabs; 22/22 (100%)	M-gene: ns H5: ns N1: ns H5-HP: ns (Figure 2h)	B1
10 2/12/21	Commercial free-range layers: 22,000	One; 25 pairs of swabs; 24/25 (96%)	M-gene: ** H5:*** N1: *** H5-HP: *** (Figure 2e)	B2
11 4/12/21	Commercial layers: 70,000	One; 24 pairs of swabs; 23/24 (96%)	M-gene: ** H5: ** N1: ** H5-HP: ** (Figure 2f)	B2
12 5/12/21	Indoor layers: 30,000	One; 20 pairs of swabs; 20/20 (100%)	M-gene: ns H5: ns N1: ns H5-HP: ns (Figure 2g)	B2
13 5/12/21	Commercial free-range layers: 96,000	One; 20 pairs of swabs; 20/20 (100%)	M-gene: ns H5-HP: ns N1: ns (Appendix A)	B2
14 5/12/21	Indoor commercial duck rearers: 40,000	One; 20 pairs of swabs; 20/20 (100%)	M-gene: ** H5-HP: *** N1: ** (Appendix A)	B1
15 6/12/21	Indoor layers: 7000	One; 22 pairs of swabs; 20/22 (91%)	M-gene: **** H5-HP: **** N1: **** (Appendix A)	B2
16 7/12/21	Commercial turkey breeders: 4000	One; 22 pairs of swabs; 22/22 (100%)	M-gene: *** H5-HP: **** N1: *** (Appendix A)	B2
17 8/12/21	Indoor commercial layers: 32,000	One; 22 pairs of swabs; 22/22 (100%)	M-gene: ns H5-HP: ns N1: ns (Appendix A)	B2
18 9/12/21	Commercial turkey rearers: 8400	One; 22 pairs of swabs; 22/22 (100%)	M-gene: **** H5-HP: **** N1: **** (Appendix A)	B2
19 10/12/21	Indoor commercial pullets: 393,600	Two; 40 pairs of swabs; 24/40 (60%) [9/20 [45%] and 15/20 [75%]]	M-gene: **** H5-HP: **** N1: **** (Appendix A)	No WGS obtained
20 11/12/21	Indoor commercial layers: 22,730	One; 22 pairs of swabs; 22/22 (100%)	M-gene: * H5-HP: * N1: * (Appendix A)	B2
21 11/12/21	Indoor commercial layers: 36,000	One; 22 pairs of swabs; 22/22 (100%)	M-gene: ns H5-HP: ns N1: ns (Appendix A)	B2
22 13/12/21	Indoor commercial ducks: 39,900	One; 20 pairs of swabs; 20/20 (100%)	M-gene: **** H5-HP: **** N1: **** (Appendix A)	B1
23 13/12/21	Indoor commercial layers: 15,500	One; 20 pairs of swabs; 18/20 (90%)	M-gene: *** H5-HP: *** N1: *** (Appendix A)	B2
24 13/12/21	Commercial broiler breeders (indoor): 14,800	One; 22 pairs of swabs; 21/22 (96%)	M-gene: ns H5-HP: ns N1: ns (Appendix A)	B2
25 14/12/21	Indoor commercial layers: 700,000	One; 22 pairs of swabs; 22/22 (100%)	M-gene: ns H5-HP: ns N1: ns (Appendix A)	B2
26 14/12/21	Indoor commercial broilers: 114,000	One; 22 pairs of swabs; 5/22 (23%)	M-gene: ns H5-HP: ns N1: ns (Appendix A)	B2
27 21/12/21	Indoor hobby ducks (Muscovy ducks): 100	One; 22 pairs of swabs; 20/22 (91%)	M-gene: **** H5-HP: **** N1: **** (Appendix A)	B1
28 27/12/21	Fattening turkeys (indoor): 30,000	One; 22 pairs of swabs; 22/22 (100%)	M-gene: **** H5-HP: **** N1: **** (Appendix A)	B2
29 27/12/21	Turkey stags: 3900	One; 22 pairs of swabs; 22/22 (100%)	M-gene: **** H5-HP: **** N1: **** (Appendix A)	B2
30 29/12/21	Finishing turkeys (in finishing houses): 3600	One; 20 pairs of swabs; 19/20 (95%)	M-gene: ** H5-HP: * N1: * (Appendix A)	B2
31 4/1/22	Fattening turkeys: 4240	One; 22 pairs of swabs; 22/22 (100%)	M-gene: **** H5-HP: **** N1: **** (Appendix A)	B2
32 12/1/22	Grandparent turkey breeders: 9500	One; 22 pairs of swabs; 22/22 (100%)	M-gene: **** H5-HP: *** N1: *** (Appendix A)	B2
33 12/1/22	Turkey breeders: 4700	One; 22 pairs of swabs; 21/22 (96%)	M-gene: **** H5-HP: **** N1: **** (Appendix A)	B2
34 3/2/22	Commercial broiler breeders: 15,000	One; 22 pairs of swabs; 22/22 (100%)	M-gene: **** H5-HP: **** N1: **** (Appendix A)	B1
35 20/2/22	Turkey rearers: 7500	One; 20 pairs of swabs; 20/20 (100%)	M-gene: **** H5-HP: **** N1: **** (Appendix A)	B2
36 20/2/22	Pheasant breeders: 6000	One; 20 pairs of swabs; 20/20 (100%)	M-gene: **** H5-HP: **** N1: *** (Appendix A)	B2
37 20/2/22	Pheasant breeders: 5000	One; 20 pairs of swabs; 20/20 (100%)	M-gene: ns H5-HP: ns N1: ns (Appendix A)	B2
38 28/2/22	Commercial fattening ducks: 60,000	One; 22 pairs of swabs; 22/22 (100%)	M-gene: **** H5-HP: **** N1: **** (Appendix A)	B2
39 10/3/22	Indoor fattening ducks: 19,200	Two; 60 pairs of swabs; 44/60 (73%) [20/30 [67%] and 24/30 [80%]]	M-gene: **** H5-HP: **** N1: * (Appendix A)	B2
40 10/3/22	Indoor fattening ducks: 700	Two; 44 pairs of swabs; 32/44 (73%) [24/24 [100%] and 8/20 [40%]]	M-gene: *** H5-HP: * N1: ** (Appendix A)	B2
41 24/3/22	Indoor layer ducks: 13,500.	Two; 60 pairs of swabs; 44/60 (73%) [18/28 [64%] and 26/32 [81%]]	M-gene: **** H5-HP: **** N1: **** (Appendix A)	B2

Table 1 footnotes provide additional information for the following six IPs: IP1: Despite small size of IP, it was included because 20 pairs of swabs were provided; IP5: Not a commercial premises, but included as the first instance of an anseriform (domestic duck) IP at the early stage of the UK epizootic, with 20 pairs of swabs provided; IP27: Not a commercial premises, but included as an instance of another anseriform (domestic duck) IP at this stage of the UK epizootic, with 22 pairs of swabs collected; IP39: Originally sampled during premovement surveillance (see main text and Appendix A), with two epidemiological units sampled at the subsequent Report Case stage; IP40: The first epidemiological unit comprised 3- and 5-week-old ducks, while the second contained 7-week-old ducks. IP40 was the only instance in this study that registered H5-seropositive ducks with the earlier clade 2.3.4.4 H5N8 antigen (Appendix A). Seven of the 24 H5N1 HPAIV-infected ducks at the first unit were coinfected with APMV-1 (Appendix A); IP41: Originally sampled during premovement surveillance (see main text and Appendix A), with two epidemiological units sampled at the subsequent Report Case stage.

## Data Availability

Available on request.

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
