# Peer review of "Efficient and Informative Laboratory Testing for Rapid Confirmation of H5N1 (Clade 2.3.4.4) High-Pathogenicity Avian Influenza Outbreaks in the United Kingdom"

_viruses, 2023, doi:10.3390/v15061344_

Round 1

Reviewer 1 Report

This article is written using a huge amount of material used for the diagnosis of highly pathogenic H5 virus in poultry . The article is of great interest to specialists in the field of virology and veterinary medicine, especially for specialists engaged in the diagnosis of highly pathogenic influenza viruses. It is written in a clear and accessible language and will be useful for poultry specialists. I have no comments, I believe that this article can be accepted for publication without changes.

Reviewer 2 Report

The study presents real-time RT-PCR results for diagnosing an H5N1 outbreak in poultry in the UK during 2021-2022. These real-world data would be highly informative in establishing efficient detection methods for HPAIV. The manuscript is well written and displays detailed results. I understand that this study focused on diagnostic results, however, data on clinical signs or mortality rates at the infected premises at the time of diagnosis could be quite informative. In the context of diagnosis, the timing of sample collection after symptom onset is very important, therefore, data on clinical signs or mortality rates could be added to Table 1. Please see my additional comments below.

Major comments

1. The authors mention that N1 RRT-PCR remains effective at the 'flock level.' What does 'flock level' mean? RRT-PCR doesn't appear to be a field diagnostic method. Please rephrase the sentence for clearer reader understanding. Additionally, as the NA subtype of clade 2.3.4.4b viruses continuously changes due to reassortment and pathogenicity is determined by the HA protein, a more thorough discussion is needed to explain why the UK retained N1 real-time PCR for HPAI testing.

2. As mentioned earlier, adding clinical sign and mortality data to Table 1 would greatly aid readers.

3. Lines 195-209: This paragraph seems inappropriate for this section. It needs to be relocated to another section.

4. Why did the authors decide that a p-value <0.0001 is highly significant? Other studies usually consider p<0.001 as highly significant, or they do not distinguish between highly significant and significant differences.

5. The results for IP 39 and 41 are duplicated in sections 3.2 and 3.3 of the results. I recommend consolidating all results for IP 39 and 41 in section 3.3.

6. The authors examined mutations in the primer-probe binding site using UK viruses isolated in this study. However, since HPAIVs spread globally and are frequently transmitted to other countries by wild birds, the primer and probe should be evaluated against global HPAIV sequences. I recommend evaluating mutations in primer-probe binding sites using more sequences from the GISAID database.

Minor comments

Line 40-41: Clade 2.3.4.4 viruses have evolved into subclades a-h, as classified by WHO, and this classification is widely used in related studies. The 2014 outbreak was caused by 2.3.4.4c, and 2.3.4.4b viruses have caused outbreaks since 2016. Thus, I recommend updating the sentences with more detailed clade classifications.

Line 50: RealTime -> real-time

Line 84: robotically -> it would be more appropriate to provide exact information, including the brand and model of the robotic machine.

Reviewer 3 Report

In this study, the authors present an effective high-throughput measurement and detection method for highly pathogenic avian influenza virus infection. The relevance of this study is underscored by the ongoing avian influenza virus panzootic, which is the longest and largest ever reported. The data generated in this study were obtained from samples taken from poultry holdings suspected or confirmed to be infected with avian influenza. These data serve as a basis for the generation and improvement of official reports.

The study describes the testing and evaluation of clinical samples taken from a series of outbreaks in poultry holdings with a high number of samples using a PCR method procedure, which provided information on the effectiveness of pooling strategies. Additionally, the study provide an overview of the genetic diversity that could result in primer mismatch and subsequently lead to false negative testing. The statistical calculations in this study are profound, and we have provided some minor remarks to further clarify the results.

remarks: 

title: The title is slighly misleading. The authors title a "testing algorithm" wetting for a bioinformatic approach, but present a comprehensive testing scheme with profound statistical evaluation. If they use any algorithm for their procedure it is needed to describe the algorithm and provide the source code. If not please retitle eg. in "testing scheme" or "testing procedure"

abbrevations: The authors do have the tendency of using a lot of abbreviations and introduce, or invent a lot. Even if the authors correctly define all abbreviations the frequent use of it and the use of uncommon abbreviation diminished readibility. The text should be scoured for non essential abbreviations. Sentences with chains of abbreviations should be rewritten. 

Table 1 The table head is to long to be informative. In addition the table includes a long footnote. The table span several pages. All this items lead to the need of redesigning the table and consideration if putting it into the supplement. 

Figure 1 shows ct values with their mean and standard deviations. The logarithmic character of ct values needs to be taken into acount. Axes need to be rescaled. Median values and quartile in a standard Whisker or box plot would be more appropriate.
